# Development of a Functional Dark Chocolate with Baobab Pulp

**DOI:** 10.3390/foods12081711

**Published:** 2023-04-20

**Authors:** Sara Monteiro, João Dias, Vanda Lourenço, Ana Partidário, Manuela Lageiro, Célia Lampreia, Jaime Fernandes, Fernando Lidon, Fernando Reboredo, Nuno Alvarenga

**Affiliations:** 1Faculdade de Ciências e Tecnologia, Campus da Caparica, Universidade Nova de Lisboa, 2829-516 Caparica, Portugal; 2Instituto Politécnico de Beja, Escola Superior Agrária, Rua Pedro Soares, 7800-295 Beja, Portugal; 3GeoBioTec Research Center, Faculdade de Ciências e Tecnologia, Campus da Caparica, Universidade Nova de Lisboa, 2829-516 Caparica, Portugal; 4Center for Mathematics and Applications (NOVA Math), Department of Mathematics, NOVA SST, Universidade Nova de Lisboa, 2829-516 Caparica, Portugal; 5UTI, Instituto Nacional de Investigação Agrária e Veterinária IP, Quinta do Marquês, 2780-157 Oeiras, Portugal

**Keywords:** chocolate, baobab, antioxidant, phenolic compound, vitamin C and minerals

## Abstract

In recent years, cocoa and dark chocolate have attracted the interest of consumers not only for their sensory characteristics but also for their nutritional properties and positive impact on health. The baobab is a fruit of African origin with a sour and slightly sweet flavour, widely consumed by local communities due to its unique nutritional features. The aim of this work was to evaluate the impact of the concentration of baobab flour in the development of functional dark chocolate, including physical, chemical, nutritional and sensory evaluations. The results presented a positive correlation between the incorporation of baobab flour and the antioxidant activity (up to 2297 mmol TE/100 g), vitamin C content (up to 49.7 mg/100 g), calcium (up to 1052 mg/kg), potassium (up to 10,175 mg/kg), phosphorus (up to 795.9 mg/kg), chlorine (up to 235.4 mg/kg) and sulphur (up to 1158 mg/kg). The sensory evaluation of dark chocolate with 3% baobab presented the highest evaluation on the parameters “texture” and “overall flavour”, while the parameter “overall flavour” presented the lowest evaluation on chocolate with 9% baobab. No influence was observed on fatty acid profile, protein, fat and hardness.

## 1. Introduction

The use of cocoa, in food or as a ritual element, is referenced among the native populations of Central America since 1800 BC [1]. Brought to Europe in the 16th century, it became a popular drink among the elites. In addition to being a beverage, cocoa has been attributed therapeutic properties and prescribed as a medicine for the treatment of various illnesses such as depression, coughs, colds, mental disorders, digestive problems and fertility problems [2]. In fact, recent studies support the positive health impacts of chocolate, especially on heart disease, blood pressure and endothelial function. Additionally, chocolate presents a high polyphenolic content, especially catechin and proanthocyanidin [3,4,5,6], high antioxidant activity, antibacterial, anti-inflammatory, anti-allergenic, anti-viral and anti-cancer properties [7].

Nowadays, chocolate is a widely distributed food in its pure form, or used in the production of a spectrum of other foods, with great consumer acceptance [8]. Some available chocolate references include freeze-dried fruit, dried nuts, seeds, olive oil [9] and sugar substitutes; however, there is still a limited offer on functional dark chocolate, particularly concerning higher polyphenolic content and antioxidant activity [10].

Many authors already consider chocolate as a functional food [11], i.e., a food that, in addition to the basic function of providing nutrients, has active compounds providing health benefits, despite discussions regarding the definition of functional food [12]. In this context, dark chocolate, due to its richness in cocoa solids up to 90%, may well be more beneficial than milk chocolate, which might have a maximum of only 50%.

Functional foods and novel foods are increasingly a trend of choice among consumers [13]. They form a set of ingredients that, due to their nutritional composition, are an asset in the formulation of new types of food products. In the case of chocolate, the addition of foods or functional ingredients can change its nutritional profile by reducing, for example, its fat content [14] or by adding new properties that raw food by itself does not present. One of these novel foods is baobab pulp flour.

The African baobab (*Adansonia digitata* L.) is part of the Malvaceae family, mostly found in sub-Saharan Africa [15] and used by local communities in the treatment of smallpox, diarrhoea and measles in addition to having a high nutritional value [16]. In fact, the recent literature on the characterization of baobab pulp indicates a low moisture level, high contents of flavonols, glycosides, dietary fibre, procyanidins [17,18,19], vitamin C, phenolic content, minerals such as calcium and potassium and antioxidant activity [15]. Since 2008, the legislative framework of the European Union authorizes the import of baobab pulp flour as a novel food ingredient and its use by the food industry [20]. Almost at the same time, the use of baobab pulp flour was authorized in the United States of America by the Food and Drug Administration for use as an ingredient in grain-based products, vegetable/fruit juices, milk products, beverages and sugar candies at levels ranging from 0.5 up to 3.8 percent [21]. Due to its richness in compounds considered beneficial to human health, some retailers of natural products label baobab as a “superfood”, a term without scientific support other than presumptive health effects. While science supports the health benefits of certain foods (quinoa, flax seed and buckwheat), coining them “superfoods”, it is clear the term is more useful for its marketing value than for providing ultimate nutrition [22].

Thus, the increased trend of baobab consumption, as well as that of ginger, turmeric and moringa among others, may in the near future elevate these foods to the same category.

Therefore, the aim of the present study was to evaluate the nutritional composition, bioactive compounds, mineral composition, fatty acid profile and sensorial acceptance of a new functional food product combining dark chocolate and different concentrations of baobab pulp flour.

## 2. Materials and Methods

### 2.1. Preparation of Chocolate Samples

The production of chocolate samples was started by melting dark chocolate 51% (CHD-R515-565, Sicao, Lebbeke-Wieze, Belgium) overnight in a tempering machine (Selmi Ghana Plus, Selmi SRL, Pollenzo, Italy) at 43 °C. Then, dark chocolate was tempered, decreasing temperature to 29.5 °C under constant agitation. Then, three batches of chocolate were prepared in which baobab pulp flour was incorporated in different concentrations (3%, 6% and 9% of baobab (*w*/*w*)) by mixing and homogenizing the baobab in the tempered dark chocolate. Commercially branded baobab of Senegalese origin (ISWARI, Setubal, Portugal) was used. A control batch was also included, i.e., dark chocolate without baobab pulp flour. Each batch was distributed in polycarbonate moulds (with 15 cavities of 7 g), followed by cooling and solidification in a refrigerator (Infrico AN10021F, Lucena, Spain) at 6 °C for 15 min. After, samples were unmoulded, packed in oriented polypropylene (OPP) film and stored at 20 °C for 20 days. The chocolate samples were prepared at Sugar Bloom/Mestre Cacau facilities (Beja, Portugal).

### 2.2. Chemical Analysis

#### 2.2.1. Moisture and Ash

Moisture content was determined at 103 °C by a gravimetric method according to Assogbadjo et al. [23]. Ash content was determined at 550 °C for 8 h by a gravimetric method according to Godočiková et al. [24]. Samples were evaluated in duplicate and expressed as a mass percentage.

#### 2.2.2. Protein

The protein content was calculated from the total nitrogen determined by Kjeldhal method, based on Roda and Lambri [25] and using 6.25 as the conversion factor according to AOAC method 970.22 [26]. Nitrogen content was determined using the digestion block and steam distillation (Tecator 2020 Digestor, Foss, Hillerød, Denmark) with colorimetric endpoint detection (Kjeltec 2300 Analyzer Unit, Foss, Denmark). Samples were evaluated in duplicate and expressed as a mass percentage.

#### 2.2.3. Fat

The extraction of fat was performed by hot solvent extraction (petroleum ether) in a Soxhlet extractor (J.P. Selecta, Barcelona, Spain). After extraction, the solvent was evaporated in a rotary evaporator with a water bath at 40 °C (Buchi Rotavapor R-114, Waterbath B-450 and Vacuum system B-169, Flawil, Switzerland). Then, the fat was placed inside an oven (Memmert, model U50, Schwabach, Germany) at 85 °C for 1 h, cooled and weighed. Samples were evaluated in duplicate and expressed as a mass percentage according to previous studies on baobab samples by Azzatul et al. [27].

#### 2.2.4. Fatty Acids

After fat determination, triglycerides contained in the extracts were converted in methyl esters following the transesterification method described in ISO 5509:2000. To 0.1 g of fat 4 mL of isooctane was added and stirred to dissolve the sample. Then, 2 mL of potassium hydroxide solution in methanol [28] was added and the mixture was stirred vigorously for about 30 s. Subsequently, using a chromatographic syringe, 2 µL of the methyl ester solution was collected and injected for chromatographic analysis. The qualitative and quantitative profile of fatty acids was determined by gas chromatography using a gas chromatograph (Trace GC 200 ThermoQuest CE instruments, Milano, Italy), a flame ionization detector and a DB-23 capillary column (50% cyanopropyl methylpolysiloxane), 60 m length, 0.25 mm i.d., 0.25 µm film thickness. The conditions for the gas chromatograph were oven temperature with programming from 150 to 220 °C at 5 °C/min, with a final time of 60 min. Injector and detector temperatures were 220 and 280 °C, respectively. Carrier gas–Helium (70 kPa) had a split flow ratio of 1/40. The fatty acids were identified based on a comparison of the retention times obtained in the sample with those obtained, under the same analytical conditions, for a reference mixture of 52 FAMES (Nu-Chek-Prep, Inc., Elysian, MN, USA). The analyses were performed in triplicate. Fatty acid quantification was performed by internal normalization, where each FAME is expressed as a percentage of the total FAMEs present in the chromatogram.

#### 2.2.5. Elemental Analysis

The elemental determination of chocolate samples, expressed in mg/kg, was performed in triplicate by using an X-ray analyser (Thermo Scientific, Niton model XL3t 950 He GOLDD+, Waltham, MA, USA) as described in [29,30]. Detection limits using the optimum mining mode for a period of 120 s under high purity helium atmosphere were: Ca = 65 mg/kg; Cl = 75 mg/kg; K = 200 mg/kg; P = 450 mg/kg; S = 90 mg/kg; Zn = 6 mg/kg. Plant reference materials used for data validation were orchard leaves (SRM 1571) and poplar leaves (GBW 07604). The recovery values were near 95%.

#### 2.2.6. Total Phenolic Content (TPC)

Methanolic extracts of the chocolates for determination of TPC were prepared based on the protocol described by Cerit et al. [31]. About 1.0 g of each sample was extracted with 20 mL of methanol (100%), followed by homogenization in a Polytron homogenizer (Ika, Ultra-Turrax T25, Staufen, Germany), placed for 5 min in an ultrasonic bath (Bransonic, Branson 5200, Branson, MO, USA) and incubated overnight at 4 °C. Mixtures were then centrifuged (4500× *g*; 4 °C; 20 min; Sigma, 2K15, Osterode am Harz, Germany). Supernatants were filtered with Whatman 41 filter paper and stored at −20 °C for further analysis. Total phenolic content in the methanolic extract was determined by spectrophotometry at 725 nm, based on published literature [32,33,34,35] using the Folin–Ciocalteu reagent and gallic acid as standard. Total phenolic content was expressed as milligrams of gallic acid equivalents per 100 g of sample (mg GAE/100 g). Samples were evaluated in triplicate.

#### 2.2.7. Antioxidant Activity (AA)

Antioxidant Activity was determined in the methanolic extracts of the samples using the DPPH reagent (2,2-diphenyl-1-picrylhydrazyl) by spectrophotometry at 580 nm, based on Brand-Williams et al. [36]. A calibration curve was prepared using trolox from 25 to 800 µM. Results were expressed as millimol of trolox equivalent per 100 g of sample (mmol TE/100 g). Samples were evaluated in triplicate.

#### 2.2.8. Vitamin C

Vitamin C content was determined by high-performance liquid chromatography, HPLC, based on Monteiro et al. [15] after extraction with metaphosphoric acid and reduction of dehydroascorbic acid to ascorbic acid with an aqueous cysteine solution (4% *w*/*v*) and trisodium phosphate solution (20% *w*/*v*). Samples for extraction of vitamin C were weighed according to the percentage of added baobab to the chocolate, namely 6 g (for 6% and 9% of added baobab (*w*/*w*)), 8 g (for 3% of added baobab) and 10 g (for control without baobab). Extractions were performed in triplicate and results were expressed as mg of ascorbic acid/100 g of sample.

### 2.3. Physical Analysis

#### 2.3.1. Texture Analysis

Texture determination was performed using a texture analyser (TA.XT Plus100, Stable Micro Systems, Godalming, UK) with 100 N load cell at 20 ± 1 °C. A 3 mm diameter aluminium cylindrical probe was used at a penetration depth of 5 mm and a test speed of 1.0 mm/s. Texture parameter hardness (maximum force in N) was calculated. Samples were evaluated ten times.

#### 2.3.2. Colour

Colour evaluation, using 10 replicates, was performed instrumentally using a Minolta CR-300R colorimeter (Minolta, Osaka, Japan), with reference to the CIE-LAB coordinate system: *L** for brightness and *a** and *b**, which define the colour between red-green and blue-yellow, respectively. The parameter whiteness index (WI) was calculated according to Equation (1) [37]:(1)$$WI=100-\sqrt{{\left(100-L*\right)}^{2}+{a*}^{2}+{b*}^{2}}$$

### 2.4. Sensory Analysis

The characterization of the sensory profile of the samples was based on the quantitative descriptive analysis described by Belščak-Cvitanović et al. [38] with some modifications. Sensory evaluation was performed with 21 untrained tasters belonging to the National Institute for Agricultural and Veterinary Research (INIAV), using a five-point hedonic scale—from “1-deslike extremely” to “5-like extremely”—to evaluate the parameters colour, texture, odour, acidic taste and overall flavour. All samples were evaluated in individual booths under white lighting and at room temperature, and samples were provided in Petri dishes coded with random 3-digit numbers.

### 2.5. Statistical Analysis

The statistical analysis of the data referring to the multiple comparisons of the constituents’ means across the different baobab concentration groups (say, G1 to G4 for 0, 3, 6 and 9% baobab concentrations) was performed using the statistical software R, version 4.1.2 (GNU General Public License, Boston, MA, USA).

In order to assess the pairwise differences in the means of the several constituents across the different concentration groups, the following statistical tests were performed:H_0_: μ_1_ = μ_2_ = μ_3_ = μ_4_ vs. H_1_: ∃ I ≠ j such that μ_i_ ≠ μ_j_,(2)
and
H_0_: μ_i_ = μ_j_ vs. H_1_: μ_i_ ≠ μ_j_,(3)
with i, j ∈ {1, 2, 3, 4} referring to the distinct concentration groups G. In particular, the hypotheses in Equation (3) were tested only when the null hypothesis in Equation (2) was rejected.

Because statistical tests usually rely on data population assumptions, these should ideally be assessed prior to the comparative analysis. In this way, for testing the hypotheses in Equations (2) and (3): (i) the F-ANOVA plus the Tukey tests, respectively, should be used in the case of data normality within concentration groups and variance homogeneity across concentration groups [39]; (ii) the F-Welch-ANOVA plus the Dunnet–Tukey–Kramer tests, respectively, should be used in the case of data normality within concentration groups but no variance homogeneity across concentration groups [40,41] and (iii) the non-parametric Kruskal–Wallis plus Dunn tests, respectively, should be used in the case of no data normality within concentration groups but variance homogeneity across concentration groups [42,43]. Due to the particularly small size of the dataset (only three observations per group) and the fact that statistical tests are highly underpowered for small sample sizes such as this one, the analysis proceeded assuming (without testing) the validation of the normality and variance homogeneity assumptions and therefore used the tests described in (i), which are themselves also underpowered (i.e., they may fail in the detection of potential true differences). All the statistical results should thus be interpreted in light of the previous comments. All samples were assumed as independent and the 5% significance level to decide against the null hypothesis was considered in all tests.

Principal component analysis (PCA) allows a dimension reduction of a large dataset of variables and establishes a correlation between samples and the variables under analysis. Principal component analysis was performed to search for possible correlation between samples and the difference that the addition of baobab flour induces in the samples produced and to understand how the variables in analyses induce this difference. For such analysis software STATISTICA 8.0 (StatSoft, Tulsa, OK, USA) was used.

## 3. Results and Discussion

The F-ANOVA test (used to test the hypotheses in Equation (2)) identified 13 constituents as having at least one significant difference between the group means. For these constituents, Tukey’s test (used to test the hypotheses in Equation (3)) identified differences between at least two groups in 12 out of the 13 constituents. In particular, constituent TPC, which was declared as having at least one significant difference in group means in Equation (2), was not declared significant in Equation (3), a fact that can be related to the lack of power of Tukey’s test for small samples. One should also acknowledge the possibility of the presence of some atypical data points (outliers) having a deleterious effect on the power of both the F-ANOVA and Tukey’s tests in detecting significant differences across concentration groups. Indeed, it takes just one outlier to spoil the sample average and standard deviation. Plus, the smaller the sample, which is the case here, the greater the effect of an outlier in the subsequent statistical analysis.

In all the tables presented throughout this section, only the constituents declared significant while testing Equation (2) re assigned with superscript letters a, b and c, which refer to the results of Tukey’s test [39] (groups with different letters meaning that the difference in means is significant at the 5% level).

### 3.1. Physical Parameters and Centesimal Composition

Table 1 presents the mean values and standard deviations for the chemical and physical parameters of chocolate samples with the incorporation of baobab.

Moisture values increased from 2.1% to 2.7% as the percentage of baobab pulp flour increased in the mixture, which is related to the hydration capacity of baobab flour [13]. Furthermore, control samples (no pulp flour added to dark chocolate) exhibited 1.6% moisture, similar to Vásquez et al. [44], in different commercial dark chocolates, and to Torres-Moreno et al. [45] when studying dark chocolates from different origins and processing conditions.

The ash content of chocolate is mainly related to the mineral elements of cocoa mass and the literature refers to the potential of dark chocolate as a significative source of essential elements to the human diet [46,47,48,49]. Our results varied between a narrow range from 1.7% to 1.9%, lower than previous works on milk chocolate with spices [50], moringa leaf powder [51] and dark chocolates from different geographical origins [49].

The levels of protein ranged between 4.8% and 5.2% and do not seem to be affected by the incorporation of baobab due to its reduced protein content [52].

Total phenolic content values ranged from 210.2 to 305.8 mg GAE/100 g, lower than those observed in previous works regarding the enrichment of chocolate with cinnamaldehyde [53], fruits/nuts [24] and stevia/peppermint leaves [38]. However, the results might well be affected by the extraction methods used in the reactive and industrial processing of cocoa beans [54], which may limit comparisons. In fact, the TPC of chocolate depends on several factors related to the raw material, processing method and storage, due to the sensitivity of some compounds to high temperatures, the presence of light, oxygen and pH [24].

The results of the antioxidant activity in chocolate samples were influenced by the incorporation of baobab flour, presenting values between 1531 mmol TE/100 g (control) and 2297 mmol TE/100 g (9% baobab). In fact, the available literature refers to the high AA values of baobab pulp flour when compared with other fruits such as oranges or strawberries [55], mainly in the hydrophilic extract [56]. Previous reports also noted a significant improvement in the antioxidant capacities of chocolate with the incorporation of dried cranberries and raisins in dark and milk chocolate [10].

The vitamin C content ranged from a value below the limit of quantification in the control sample to 49.7 mg ascorbic acid/100 g (9% baobab), proportional to the incorporation of baobab flour in dark chocolate, a consequence of the high levels of vitamin C in baobab flour [55], about 3 to 7 times higher than in orange [57,58]. However, other studies involving chocolate with added moringa leaf powder [59] presented values ranging between 149.3 mg/100 g and 201.4 mg/100 g, while [60] there were observed values of 122 mg/100 g in chocolate enriched with guava powder.

The hardness values were not influenced by the different percentages of baobab pulp flour, with levels similar to previous results on plain dark chocolate [38,61] and dark chocolates with the substitution of sucrose by a combination of D-tagatose and inulin [62]. According to the available literature, hardness is correlated with the particle size distribution of solids [63,64,65]. Colour is a fundamental criterion for the food industry and is the basis for many consumer purchasing choices [48]. The chocolate industry is no exception and is commonly used as a quality parameter [66,67,68]. The literature refers to “fat bloom” as a change in surface colour, or loss of gloss, shifting into a grey/white appearance of the chocolate surface [69]. Possible causes may be improper tempering, large fluctuations in storage temperature [37], or fat migration in the case of filled chocolates [70]. The obtained results for the L parameter ranged between 27.6 (control) and 28.1 (9% baobab), with no significant differences. However, both colour parameters *a** and *b** presented an influence due to the incorporation of baobab: colour parameter *a** increased from 5.3 (control) to 6.4 (6% baobab), increasing the intensity of red colour, while parameter *b* increased from 5.5 (control) to 6.3 (6% baobab), increasing the intensity of yellow colour. Despite the differences between the control and samples with baobab, colour parameters were not affected, which agrees with previous results on chocolate [53,71], where an increase in *a** and *b** parameters was also observed on surface colour, resulting in a higher intensity of red and yellow. Parameter WI, calculated from lab values, varied between 27.1 (6% baobab) and 27.5 (9% baobab), with no significant differences at a 0.05 significance level. The observed values, similar to previous works with plain dark chocolate [66,68,72], do not reveal changes in the colour surface due to the blooming process, possibly due to the short-term storage, around 20 days.

The results show significant differences between the samples with different percentages of baobab in relation to the parameters *a**, *b**, AA, moisture, ash and vitamin C. However, the hardness, *L**, WI, fat and protein showed no significant differences.

### 3.2. Fat and Fatty Acids

The fat content (Table 1) was not affected by the incorporation of baobab, a consequence of its low-fat content [52]; the results ranged between 30.2% and 37.8%, similar to studies conducted on chocolate bars enriched with chia seeds [73], but lower than chocolate enriched with passion fruit seeds and orange peel [74].

Table 2 presents the fatty acid profile of the control and samples with different percentages of baobab. The values for total saturated fatty acids (SFA), total monounsaturated fatty acids (MUFA) and total polyunsaturated fatty acids (PUFA) are also presented. All samples revealed similar profiles and no significant differences were observed with the exception of C20:1 (*cis* 11), higher in samples with 6% and 9% baobab (Table 2). The content of SFA ranged from 61.24% (6% baobab) to 62.41% (control), with the predominance of palmitic and stearic acids similar to the available literature [75,76,77]. On the other hand, oleic acid presented the highest values among MUFA, between 31.68% (control) and 33.00% (6% baobab). The similarities observed are the consequences of the reduced fat content of baobab flour [20].

The presence of saturated fatty acids in foods—such as myristic, palmitic and stearic—is quite relevant in nutritional terms due to their negative impact on health. According to the literature, myristic acid represents the SFA with the highest capacity for atheroma formation and has four times more cholesterol-raising effect than palmitic acid [78]; the presence of stearic acid increases cardiac risk more than palmitic and myristic acids [79]. On the contrary, oleic acid is considered responsible for lowering cholesterol levels and presents preventive effects (effect) on several chronic diseases such as cardiovascular diseases, cancer and age-related cognitive decline [45]. Linoleic acid was the main PUFA, presenting values around 3%, followed by linolenic acid, around 0.2% (Table 2). However, in reduced amounts, when compared to SFA and MUFA, the daily consumption of these two fatty acids has a protective effect against cardiovascular disease, cancer and diabetes [80,81].

### 3.3. Elemental Analysis

The results of the elemental analysis are shown in Table 3. Overall, mineral content was affected by the concentration of baobab pulp flour on chocolate samples except Zn, where no significant differences were observed.

Potassium (K) presented the highest mineral content, ranging from 8058 mg/kg (control) to 10175 mg/kg (9% baobab), higher than reported values for plain dark chocolate [44]. Potassium participates in maintaining normal cellular functions limiting blood pressure (a consequence of an excessive sodium intake) and reducing the occurrence of kidney stones [82]. Sulphur (S) ranged from 988.2 mg/kg (control) to 1158 mg/kg (9% baobab); thus, the incorporation of baobab in chocolate samples does not seem to affect S levels. Sulphur has biological importance due to its integration into many molecules including amino acids, proteins, enzymes and vitamins [83]. After calcium and phosphorus, sulphur is the third most abundant mineral in the human body and is mostly provided by sulphur-containing amino acids [84]. Calcium (Ca) ranged from 648.9 mg/kg (control) to 1052 mg/kg (9% baobab), similar to previous reports on plain dark chocolate [46]. Calcium is the element with the highest concentration in the human body and is required for muscle contraction, nerve impulse transmission, hormone secretion and the formation of bones and teeth [48]. Phosphorus (P) ranged from 667.9 mg/kg (control) to 795.9 mg/kg (9% baobab). Phosphorus is present in several tissues of the human body, for example, in the cell membrane, in oxygen transport and in bone mineralization [85]. Phosphorus is a major constituent of the structural components of skeletal tissues [86,87]. Other mineral elements present in chocolate samples include chlorine (Cl), between 79.83 mg/kg and 235.4 mg/kg, and zinc (Zn), around 18 to 22 mg/kg and similar to reported values for plain dark chocolate [46]. Zinc plays a role in the immune system and its lack causes the atrophy of the lymphoid organs [88], which may represent a public health problem in developing countries [89]. Chlorine is one of the most important elements for the proper functioning of the human body, regulating the osmotic pressure and acid-base balance and helping maintain blood pH at appropriate values [90]. In addition, chlorine helps to keep the amount of fluid in and out of cells in balance in the body. It also contributes to proper blood volume and blood pressure, as well as being one of the components of gastric juice, helping in the digestion of food [91].

### 3.4. Principal Component Analysis

Principal component analysis (PCA) was carried out (Table 4) in order to perform the integration of results from different parameters using the main nutritional attributes (ash, total polyphenolic content, fat content, antioxidant activity, vitamin C and the minerals calcium (Ca), potassium (K) and phosphorus (P)).

The similarity map defined by the first two principal components accounted for 88.8% of the total variance. The first component (PC1) condensed 75.1% and the second component (PC2) represented 13.7% of the total variance. The first component was correlated negatively with AA, TPC, ash, vitamin C and minerals (Ca, P and K) and positively with fat. The second component was correlated only with mineral P. Figure 1 shows the projection of the samples onto the PC1/PC2 plane.

With the principal component analysis it was possible to perform the integration of results from different variables and it was possible to highlight the following findings:Here is a clear separation between the control and the samples produced according to the percentage of incorporation of the baobab pulp flour in the chocolate.The incorporation of baobab in chocolate decreases the correlation of the samples produced with the fat variable, evidenced by the tendency of negative correlation in PC1, and proves the decrease in fat in the samples.There is a strong correlation between K, vitamin C, AA, Ca, ash and TPC that, due to its projection in PC1, shows the correlation between this set of variables and the incorporation of baobab in chocolate.

The correlation between the incorporation of baobab and the variables with relevance to the evaluation of the functional capacity of the baobab, namely K, vitamin C, AA, Ca and TPC, is clear; the increase in incorporated baobab turns dark chocolate into a potential functional food with the ability to suppress specific nutritional needs.

### 3.5. Sensory Analysis

The sensory analysis of sample chocolates was performed through a hedonic evaluation comparing chocolates with different baobab concentrations and a control sample. The results are shown in Figure 2.

Results around 4 to 4.5 are noted for “odour” and around 3 to 3.5 for “acidic taste”, not influenced by the concentration of baobab. On the other hand, the control sample presented the highest acceptance on the “color” parameter, possibly due to a colour change caused by the incorporation of baobab flour and previously identified on colour parameters *a** and *b** (Table 1). Chocolate samples with 3% baobab presented the highest values for the parameters “texture” and “overall flavour” although hardness presented no differences (Table 1). On the other hand, the sample with 9% baobab flour showed the lowest acceptability in the parameter “overall flavour”.

## 4. Conclusions

The baobab fruit is widely used by local communities in Africa, mostly as a foodstuff due to its high nutritional value; however, applications outside Africa are still scarce. An increase in the antioxidant activity and vitamin C of chocolates with the incorporation of baobab flour, especially at a concentration of 9%, was observed. The incorporation of baobab flour triggers the possibility of nutritional enhancement compared with conventional dark chocolates, namely in Ca and K. On the other hand, the fatty acid profile was not affected; the consequence of the reduced lipid content of baobab flour is insufficient to promote changes in the final product. The only exception was C20:1 (*cis* 11), higher in 6% and 9% baobab. Principal component analysis supports the cause–effect relationship between the incorporation of bobab in chocolate and the variables of interest in functional foods, especially P, Ca, vitamin C, AA and TPC. Finally, the sensory evaluation presented similar results on the parameters “odour” and “acidic taste”. Chocolate with 3% baobab presented higher classifications on the parameters “texture” and “overall flavour”, while chocolate with 9% baobab presented the lowest classification on “overall flavour”.

In conclusion, the use of baobab flour could be a valid option in the development of new chocolate products, combining the beneficial characteristics of these two ingredients.

## Figures and Tables

**Figure 1 foods-12-01711-f001:**
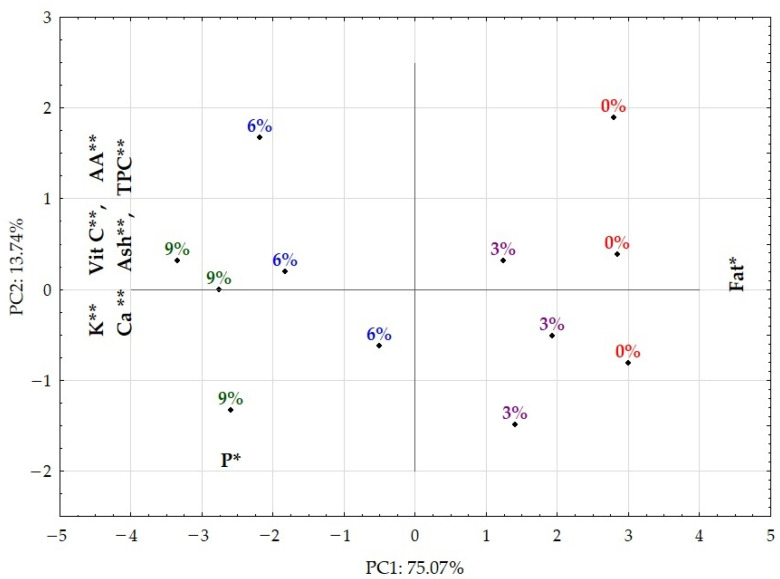
Principal component analysis: PC1 vs. PC2 projection of samples (n = 3). 0%: sample without baobab; 3%: sample with 3% of baobab; 6%: sample with 6% of baobab; 9%: sample with 9% of baobab; the most important variables for the definition of the two components are shown in each axis, indicating the direction in which each variable grows. * Moderately significant correlation values between the variable and the PC; ** strongly significant correlation values between the variable and the PC.

**Figure 2 foods-12-01711-f002:**
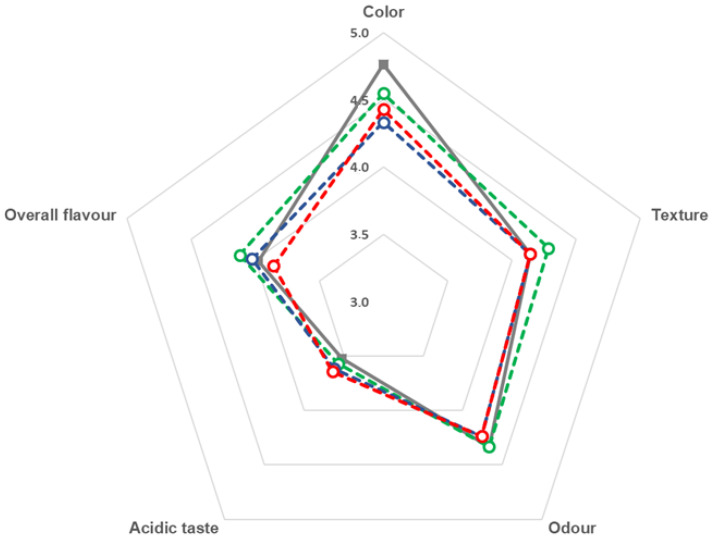
Average values of sensory analysis: control (
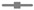
) and samples with incorporation of 3% (
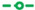
), 6% (
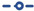
) and 9% (
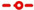
) of baobab pulp flour.

**Table 1 foods-12-01711-t001:** Mean values and the standard deviations (within brackets) of control and chocolates with baobab.

Parameter	Control	Baobab	Incorporation of Baobab (*w*/*w*)		
			3%	6%	9%
Moisture % (*w*/*w*)	1.6 (0.0) ^e^	13.31 (0.4) ^d^	2.1 (0.0) ^c^	2.3 (0.0) ^b^	2.7 (0.1) ^a^
Ashes % (*w*/*w*)	1.7 (0.0) ^c^	4.7 (0.1) ^a^	1.7 (0.0) ^c^	1.9 (0.1) ^b^	1.9 (0.0) ^b^
Protein % (*w*/*w*)	5.2 (0.1) ^a^	2.27 (0.0) ^b^	4.9 (0.1) ^a^	4.8 (0.2) ^a^	5.0 (0.3) ^a^
Fat % (*w*/*w*)	37.8 (0.2) ^a^	0.5 (0.02) ^b^	37.2 (0.8) ^a^	30.2 (9.1) ^a^	31.2 (5.2) ^a^
TPC (mg GAE/100 g)	210.2 (48.8) ^a^	460.5 (88) ^b^	228.8 (39.6) ^a^	303.4 (38.5) ^a^	305.8 (24.0) ^a^
AA (mmol TE/100 g)	1531 (285) ^b^	539.9 (161) ^c^	1676(242) ^ab^	2254 (209) ^a^	2297 (136) ^a^
Vitamin C (mg ascorbic acid/100 g)	<LQ	288.9 (37.5) ^a^	9.8 (3.3) ^d^	30.8 (4.0) ^c^	49.7 (2.5) ^b^
Hardness (N)	21.6 (0.2)	-	22.8 (1.8)	22.8 (1.5)	21.6 (1.9)
*L**	27.6 (0.2)	-	27.9 (0.2)	27.6 (0.2)	28.1 (0.3)
*a**	5.3 (0.2) ^b^	-	6.1 (0.1) ^a^	6.4 (0.3) ^a^	6.3 (0.1) ^a^
*b**	5.5 (0.2) ^b^	-	6.1 (0.1) ^ab^	6.3 (0.4) ^a^	6.1 (0.1) ^a^
WI	27.2 (0.2)	-	27.4 (0.2)	27.1 (0.3)	27.5 (0.3)

Significant differences are marked with different letters in each row; TPC: Total phenolic content; AA: Antioxidant activity; WI: Whiteness index; LQ: Limit of quantification.

**Table 2 foods-12-01711-t002:** Average values (n = 3) and standard deviations (within brackets) for the percentage of fatty acids from control and from chocolates with baobab.

Fatty Acid	Control	Incorporation of Baobab (*w*/*w*)		
		3%	6%	9%
Lauric—C12	0.05 (0.01)	0.07 (0.02)	0.06 (0.03)	0.08 (0.02)
Mystiric—C14	0.29 (0.08)	0.29 (0.11)	0.27 (0.17)	0.37 (0.11)
Pentadecanoic—C15	0.05 (0.02)	0.06 (0.02)	0.06 (0.03)	0.07 (0.02)
Palmitic—C16	25.74 (1.37)	24.90 (0.29)	24.29 (0.16)	24.34 (0.55)
Palmitoleic—C16:1 (*cis* 9)	0.31 (0.06)	0.37 (0.06)	0.39 (0.10)	0.39 (0.08)
Margaric—C17	0.55 (0.57)	0.47 (0.31)	0.42 (0.10)	0.49 (0.05)
Heptadecenoic—C17:1 (*cis* 10)	0.03 (0.00)	0.06 (0.03)	0.06 (0.02)	0.07 (0.03)
Stearic—C18	34.72 (1.10)	34.52 (0.38)	35.09 (0.78)	34.97 (0.68)
Oleic—C18:1 (*cis* 9)	31.68 (0.29)	32.87 (0.43)	33.00 (0.41)	32.74 (0.81)
Linoleic—C18:2 (*cis* 9,12)	3.39 (0.38)	3.28 (0.14)	3.13 (0.06)	3.17 (0.07)
Linolenic—C18:3 (*cis* 9,12,15)	0.23 (0.05)	0.21 (0.01)	0.21 (0.01)	0.22 (0.02)
Arachidic—C20	1.01 (0.12)	0.97 (0.06)	1.04 (0.07)	0.96 (0.04)
Gadoleic—C20:1 (*cis* 11)	0.10 (0.04) ^ab^	0.03 (0.00) ^b^	0.21 (0.08) ^a^	0.23 (0.09) ^a^
SFA	62.41 (0.04)	61.28 (0.37)	61.24 (0.58)	61.27 (0.95)
MUFA	32.09 (0.37)	33.26 (0.38)	33.60 (0.52)	33.36 (0.94)
PUFA	3.63 (0.41)	3.49 (0.13)	3.34 (0.05)	3.38 (0.09)

Significant differences are marked with different letters in each row; SFA: Saturated fatty acids; MUFA: Monounsaturated fatty acids; PUFA: Polyunsaturated fatty acids.

**Table 3 foods-12-01711-t003:** Average values and standard deviations (within brackets) of minerals (in mg/kg) of control and chocolates with baobab.

Mineral	Control	Baobab	Incorporation of Baobab (*w*/*w*)		
			3%	6%	9%
Zn (mg/kg)	21.50 (1.84)	<LQ	18.79 (0.46)	19.71 (1.52)	18.09 (1.21)
Ca (mg/kg)	648.9 (78.1) ^d^	2937 (352) ^a^	828.7 (13.1) ^c^	927.0 (29.7) ^c^	1052 (11) ^b^
K (mg/kg)	8058 (42) ^e^	40973 (340) ^a^	8693 (170) ^d^	9419 (222) ^c^	10175 (265) ^b^
P (mg/kg)	667.9 (66.2) ^c^	9273 (179) ^a^	746.7 (39.7) ^bc^	766.5 (29.8) ^bc^	795.9 (38.2) ^b^
Cl (mg/kg)	79.83 (11.35) ^e^	31997 (528) ^a^	152.7 (2.9) ^d^	193.5 (15.0) ^c^	235.4 (4.9) ^b^
S (mg/kg)	988.2 (39.9) ^c^	1675 (324) ^a^	1140 (21) ^b^	1130 (29) ^b^	1158 (39) ^b^

Significant differences are marked with different letters in each row; LQ: Limit of quantification.

**Table 4 foods-12-01711-t004:** Correlation coefficients between attributes and PC1 and PC2.

Component	PC1	PC2
Antioxidant activity (AA)	−0.91 **	0.35
Total polyphenolic content (TPC)	−0.83 **	0.4
Ash	−0.91 **	0.05
Fat content	0.67 *	−0.44
Vitamin C	−0.98 **	−0.08
Ca	−0.93 **	−0.28
K	−0.96 **	−0.19
P	−0.66 *	−0.71 **
Eigenvalue	6.01	1.10
% Variance	75.07	13.74
% Cumulative variance	75.07	88.81

* Marked values were considered moderately correlated with the PC; ** marked values were considered strongly correlated with the PC following the classification used previously [92].

## Data Availability

The data presented in this study are available on request from the corresponding author.

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
