# Peer review of "Development of a Functional Dark Chocolate with Baobab Pulp"

_foods, 2023, doi:10.3390/foods12081711_

Round 1
Reviewer 1 Report
Review foods-2249795
Why did the authors apply the term ‘functional food’ to indicate this chocolate combined with baobab pulp? Wheat were the functions related to this food item?
Section 2.1: it was not clear whether certain amount of chocolate was replaced or the baobab pulp was added into the chocolate directly without replacing the corresponding chocolate portion?
Then the authors should consider whether the term ‘incorporation’ should be replaced with ‘replacement’.
Table 3.1: The parameter ‘Fat %’ needs double checked. If the baobab contained less or more fat that the same amount of chocolate, the replacement groups should show the same trend with more replacement. The results of groups 6% and 9% incorporation were pretty strange.
Table 3.2: further discussion on the results is needed. For instance, what does it mean with different fatty acids composition? For food industry, it might indicate the antioxidant mode/effects might be different. For instance, LWT - Food Science and Technology, 118, 108737.
Line 399-400: hedonic scale evaluation does not apply for trained panelist sensory evaluation. The authors should carefully evaluate the methods and the results.
Figure 3.1: the method section introduced that the evaluation results were recorded as 1 to 5, pending on different standards/characteristics. In Figure 3.1, why did the results were labeled 0.3-0.5? In addition, why did two different places labeled as 0.5 for the parameter ‘Color’?
The current discussion focused one parameter after another. However, the authors need to pay more attention the integration of results from different parameters. They should be related to each other rather than independently. Thus, it is critical to discuss why the incorporation of baobab pulp caused the changes of various parameters systematically? A diagram or similar approach might be helpful. For instance, Food Hydrocolloids, 99, 105317; Food Hydrocolloids, 75, 164-173.
Line 34: the authors stated rheological property. However, there were no corresponding results showed related to this property. Therefore, it is inappropriate to use this term here.
Reviewer 2 Report
The authors have presented the results of a suitably designed study. The experimental plan is appropriate. However, the Results and Discussion section has not been structured well. It merely states the trend of results (increase/decrease/no significant change) without technical justifications and comparison with the existing literature base. Without an elaborate discussion of the inferences from the study, the results remain a matter of obviousness. The references cited in the discussion are not directly relevant to this study. The rationale and significance of incorporating baobab pulp in the chocolate formulation should have been emphasized more clearly in the introduction section rather than describing the history of cocoa beans and chocolate, which is well-known. The Introduction appears to be part of a review article rather than a research article wherein the background, objectives, hypothesis, and problem statement should be set out clearly. Likewise, the conclusion section is very similar to the abstract and does not state the salient findings of the study. There are glaring errors (unit of ID and film thickness of the GC capillary column) and missing information (ex. model of the Soxhlet extractor) in the materials and methods section.
Reviewer 3 Report
The current paper shows that a baobab flour can be added to a dark chocolate at 3-9% levels without significantly affecting its nutrition content, texture, odor and overall flavor. The conclusions are based on tests about the fatty acid composition in control sample and in the chocolates prepared with baobab, the elemental analysis of the four different samples, as well as on sensory analysis performed with 21 trained tasters. The paper is written in a very clear way and all methods used are explained in details.
I have the following comments and questions, which I found important to be addressed:
- An elemental analysis and fatty acid composition for the baobab flour incorporated in the chocolate is currently missing. Hence, some of the data presented within Tables 3.1 and 3.3 are difficult to understand. For example – the TPC content in the control sample is 210.2 (mg GAE/100g), it becomes 228.8 when 3% baobab flour is added, but increases to 305.8 when 9% baobab flour presents.
- One of the main features of the chocolate is that it melts at temperatures close to the body temperature. Have you considered whether the melting profile of chocolate changes when the baobab flour is added?
- How does the energy value of the chocolate changes when baobab flour is added?
- Line 115 – please explain what is the meaning of the “conversion factor” or add appropriate reference.
- Table 3.1, last 4 rows – please define L(-), a (-), b(-) and WI (-)
Round 2
Reviewer 1 Report
Review foods-2249795
The authors have addressed the questions quite well. The revised manuscript has been improved significantly. There is no further comment. The current version is acceptable for publication.
Author Response
The authors agree and are very grateful for the answer of the reviewer's 1